# Lipomatous Tumors: A Comparison of MRI-Reported Diagnosis with Histological Diagnosis

**DOI:** 10.3390/diagnostics12051281

**Published:** 2022-05-21

**Authors:** Tobias M. Ballhause, Alexander Korthaus, Martin Jahnke, Karl-Heinz Frosch, Jin Yamamura, Tobias Dust, Carsten W. Schlickewei, Matthias H. Priemel

**Affiliations:** 1Department of Trauma and Orthopedic Surgery, University Medical Center Hamburg-Eppendorf, 20246 Hamburg, Germany; a.korthaus@uke.de (A.K.); martin.jahnke@klinikum-lueneburg.de (M.J.); k.frosch@uke.de (K.-H.F.); t.dust@uke.de (T.D.); c.schlickewei@uke.de (C.W.S.); priemel@uke.de (M.H.P.); 2Department of Trauma Surgery, Orthopedics and Sports Traumatology, BG Hospital Hamburg, 21033 Hamburg, Germany; 3Department of Diagnostic and Interventional Radiology and Nuclear Medicine, University Medical Center Hamburg-Eppendorf, 20246 Hamburg, Germany; j.yamamura@uke.de

**Keywords:** lipoma, liposarcoma, sarcoma, soft tissue tumor, atypical lipomatous tumor, MRI

## Abstract

Lipomatous tumors are among the most common soft tissue tumors (STTs). Magnetic resonance imaging (MRI) is a state-of-the-art diagnostic tool used to differentiate and characterize STTs. Radiological misjudgment can lead to incorrect treatment. This was a single-center retrospective study. Two hundred and forty lipomatous tumors were included. MRI diagnoses were categorized as benign, intermediate, or malignant and were compared with histological diagnoses. Tumor volumes were measured by MRI and from surgical specimens. The tumor was correctly categorized 73.3% of the time. A total of 21.7% of tumors were categorized as more malignant in MRI reports than they were by histology, and vice versa for 5.0% of tumors. Volume measured by MRI was not different from actual tumor size in pathology. Atypical lipomatous tumors (ALTs) and liposarcomas (LPSs) were larger when compared with lipomata and occurred in older patients. Based on the MRI-suspected tumor entity, surgical treatment can be planned. Large lipomatous tumors in elderly patients are more likely to be ALTs. However, a safe threshold size or volume for ALTs cannot be determined.

## 1. Introduction

Lumps and soft tissue tumors (STTs) are frequent reasons for consulting a physician. Most soft tissue tumors are benign, and lumps are not always associated with tumors. Lipomatous tumors are the most common STTs [1]. Adipocytic tumors range from benign fat-containing masses to highly malignant liposarcomas (LPSs). Benign adipocytic tumors include lipoma, lipoblastoma, myolipoma, chondroid lipoma, spindle cell lipoma, lipomatosis, and hibernoma [2]. Mesenchymal elements can occasionally be found within lipomata. Depending on the tissue’s name, these lipomata are dubbed fibro lipoma or myolipoma as examples [3]. Most lipomatous STTs are benign, and soft tissue sarcomas are fairly rare tumors, with an incidence of 4.7/100,000 per capita [4]. Altogether, soft tissue sarcomas account for only 1% of all malignant tumors [5]. 

The first step in diagnostic procedures should be a thorough clinical assessment of the lump, followed by medical imaging. Magnetic resonance imaging (MRI) is the most common advanced imaging modality for soft tissue disorders [6,7]. Therefore, it is recommended by the European Society of Musculoskeletal Radiology (ESSR) [8]. Due to the low incidence of soft tissue sarcomas and their similar appearance in MRI, it is difficult for radiologists to differentiate them from benign lipomatous tumors [9]. Matters are complicated further by the fact that more than 70 subtypes of soft tissue sarcomas are known [10].

The main MRI characteristic that helps to distinguish between a lipoma and an atypical lipomatous tumor (ALT) is the broader and more nodular fibrous septa in the latter [11]. Homogeneity of the signal, especially in sequences with fat suppression, is an indicator of lipomata [12]. ALTs are characterized as intermediate tumors (regarding their malignancy); they grow locally (destructive) and typically do not metastasize but have the potential to further dedifferentiate [13]. This greatly differentiates ALTs from LPSs, which are malignant [2]. LPSs are composed of malignant lipoblasts with fat in their cytoplasm [3]. LPSs can further be classified into four subtypes based on their histological appearance: dedifferentiated (DDLPS), myxoid LPS (MXLPS), pleomorphic LPS (PMLPS), and not otherwise specified liposarcoma (NOSLPS) [14]. Round-cell LPSs are currently regarded as a subtype of MXLPS and are therefore included in this group [15].

In this study, we aimed to analyze the accuracy in diagnostics of MRI reports when compared with the matching histological reports of the tumors, with emphasis on their prognosticated malignancy. It is the largest contemporary study of this kind, with 240 lipomatous tumors included. 

## 2. Materials and Methods

This study has a retrospective, single-center design. Institutional Review Board approval was given. From 2011 to 2020, patients who were surgically treated at our institution were analyzed. 

MRI with a focus on soft tissue should be supported by the application of gadolinium [16]. The enrichment of gadolinium is of great help for the differentiation of STTs [17]. Only 17 MRIs in our study were without a contrast medium. The majority (*n* = 223) was performed after the application of gadolinium. Indication for open biopsy or resection and planning of the resection were completed by two senior surgeons, with both having more than ten years of experience in musculoskeletal tumor surgery. All MRIs were reported by fellowship-trained radiologists.

The primary radiologic diagnosis was compared with the final histologic diagnosis. The tumors were classified by the authors as benign (I), intermediate (II), or malignant (III). This classification was used for the radiologic and histologic results. In some cases, the radiologist described a tumor as “unclear, a biopsy is recommended”, and these were categorized as intermediate (II). Hence, a diagnosis was regarded as correct when the classification of the tumor was alike in the radiologic and histologic diagnoses. All histologic analyses were performed in the institution’s pathology department, and in cases of doubt, a reference pathology center was involved. As a standard operating procedure, all specimens of lipomatous tumors were tested for the *Mouse Double Minute 2 (MDM2)* gene.

Tumor volume was calculated by using the reported measurements in the MRI report. Statistical analysis was performed using GraphPad Software 9 (Los Angeles, CA, USA). Parametric distributions were tested with the Shapiro–Wilk test. Non-parametrically distributed data were analyzed via the Kruskal–Wallis test. Parametrically distributed data were assessed with a two-way ANOVA using Tukey’s multiple comparisons test. For all tests, a confidence interval of 95% was chosen.

## 3. Results

Altogether, 240 lipomatous tumors were included in the final analysis (Figure 1). For analysis, these were distinguished as a lipoma, an ALT, or a sarcoma (Figure 2). The occurrence of lipomata was alike in women (*n* = 60) and men (*n* = 56). Most lipomata were found in the neck/shoulder region (*n* = 35), followed by the pelvis/thigh region (*n* = 25). A total of 19 lipomata were located on the upper arm, 15 in the lower arm, and 16 in the torso. Only two lipomata were close to the knee joint, and four were in the lower leg/foot. At the time of surgery, patients with lipomata had an average age of 51.8 ± 14.0 years (Figure 3). The average volume of lipomata measured by MRI was 309.8 ± 439.8 cm^3^, compared with 353.8 ± 975.3 cm^3^ by pathology assessment. The difference in volume was not statistically significant (Figure 4). 

Slightly more men than women were affected by ALTs (Figure 2B). Most frequently, ALTs were found in the pelvis/thigh region (*n* = 31). All other body regions were less affected, with one ALT in the shoulder/neck region, one in the upper arm, and one in the lower arm. In this cohort, no ALTs were found in the knee, and only two patients had an ALT in the lower leg. At the time of surgery, patients with ALTs had a mean age of 66.0 ± 11.5 years. Hence, they were significantly older than patients with lipomata (*p* < 0.0001). The average volumes measured by MRI and by the pathologist were 1305.7 ± 1226.2 cm^3^ and 1377.7 ± 1365.7 cm^3^, respectively (Figure 4). No statistical difference in ALT volume was detected. However, the average ALT volume was significantly larger in both the MRI and the measurement after resection groups (both *p* < 0.0001). 

An LPS was diagnosed in 68 patients, with almost twice as many in men as in women (Figure 2C). Similarly to ALTs, most tumors were located in the pelvis/thigh region (*n* = 36). Moreover, two were found in the shoulder/neck region and six in the upper arm. In contrast, three LPSs were found in the lower arm and six in the torso. In two cases, the sarcoma was found in the knee region, and the lower leg/foot was affected in ten cases. On average, patients with sarcomas were 58.8 ± 17.5 years old. Therefore, they were significantly older than patients with lipomata (*p* = 0.007) and significantly younger than patients with ALTs (*p* = 0.04) (Figure 3). Average LPS volume was 681.5 ± 1148.3 cm^3^ when measured by MRI and 882.3 ± 1205.1 cm^3^ when measured in pathology. No statistical difference in volume was found when comparing MRI measurements with measurements in pathology. LPSs were significantly smaller than ALTs by MRI measurement (*p* = 0.011), but a statistical difference was not found between volumes measured in pathology.

The radiologic and histologic categories were alike in 176 cases and are regarded as correct radiologic diagnoses. An over-diagnosis by the radiologist occurred in 52 cases (Appendix A). Twelve tumors were under-diagnosed by MRI. Of the 12 falsely diagnosed tumors, 4 were intermediate, and 8 were malignant lipomatous tumors (Table 1). Tumors were under-diagnosed in six women and six men, with an average age of 63.0 ± 13.6 years. The mean tumor volume of the eight malignant tumors measured by MRI was 139.9 ± 280.6 cm^3^, and in pathology was 230.7 ± 390.9 cm^3^. Two tumors were in the neck/shoulder region, one in the lower arm, three in the truncus, three in the pelvis/thigh, and three in the lower leg/foot. All three in the lower leg/foot were malignant (Table 1).

## 4. Discussion

A reported finding by a radiologist is of utmost importance for the further treatment of a patient. After clinical examination of the swelling and the conducting of an MRI, the radiologist’s assessment frequently determines to which specialist a general practitioner refers the patient. The speed of the diagnostic process is especially crucial in malignant STTs [18]. However, due to the overall rareness of malignant STTs, this decision is an eminently difficult one [19]. Therefore, it can only be emphasized that in cases of uncertainty, the patient is referred to a sarcoma center, where there is great expertise in STTs. Evidence shows that referral to a specialized sarcoma center improves treatment and consequently increases survival [20,21]. Early diagnosis of the sarcoma increases the chances of survival and reduces the magnitude of surgery [18,22]. 

MR imaging of an STT provides the physician with essential information about the size and entity of a tumor. Improved medical imaging, advanced histological assessment, and careful tumor resection help improve reconstructive options and preserve limbs [23]. In our study, a tumor was correctly categorized by the radiologists in 73.3% of cases as benign, intermediate, or malignant. Furthermore, the MRI shows possible infiltration of the fascia, nerve structures, or blood vessels. Coran et al. reported a 100% rate of correct detection of a malignant LPS in 54 cases diagnosed by MRI. Despite the two-thirds prognostic correctness on the tumor’s malignancy in radiologic reports in our study, 12 of 240 lipomatous tumors were “underrated”. In relation to the total number of tumors, this is 5.0%. However, 8 of these 12 under-diagnosed tumors were LPSs. Every wrong diagnosis has extreme consequences for the individual patient. 

In contrast, there seems to be a trend to overestimate the malignance of lipomatous tumors [12]. In several cases, the correct diagnosis of the tumor was mentioned in the MRI report on diagnostic findings, but an additional sentence was added that malignant neoplasia could not be excluded. This might be added by radiologists for forensic reasons. We categorized these tumors in our classification system for this article as (II) intermediate. 

The final decision about the treatment of the presented soft tissue swelling lies in the hands of the attending surgeon. Due to the rare incidence of soft tissue sarcomas, great expertise is demanded from the radiologist. In cases of doubt, a second radiologic assessment might be helpful by a “reference radiology”, as this is already a standard procedure in diagnostic pathology. 

A simple “over-diagnosis” of a benign or intermediate tumor can lead to prolonged treatment. Perhaps a biopsy is favored, which might not have been necessary in the first place. Therefore, the patient receives two surgeries and perhaps two rounds of general anesthesia. 

Nevertheless, the recommendation is to biopsy all malignant and unclear tumors prior to resection [24]. This allows for a multimodal therapy approach [25]. A biopsy before resection of malignant STTs results in improved overall survival [26,27]. These findings emphasize the importance of preoperative diagnostics to estimate the tumor’s entity as accurately as possible. Brisson et al. reported a positive predicted value for ALTs of 47%. The morphological overlap with lipomata is problematic [28]. The safest discrimination between an ALT and a lipoma can be achieved by immunohistochemistry against the proteins of the *MDM2* and *CDK4 genes* [29]. ALTs can exclusively be treated with excision since they do not typically metastasize [30]. The tendency of metastization seems to depend on the locality of the ALT. Marginal excision for ALTs is favored in our institution. Size matters in ALTs. In 1994, Gelineck stated that a lipomatous lesion measured as >5 cm by MRI is highly suggestive of an ALT [31]. This was confirmed by Datir et al. in an MRI-based study of 571 patients with suspected soft-tissue neoplasms [32]. However, we could not find a threshold in volume to clearly differentiate an ALT from a lipoma by sizing. The smallest ALT in our cohort measured 60.45 cm^3^.

In our study, most liposarcomas were located in the pelvis/thigh region. This is in accordance with other findings [19,33]. LPSs have a peak age of incidence between the fifth and seventh decade [19]. There is a tendency for a higher frequency of dedifferentiated liposarcomas in older patients. Similar to our findings, Evans et al. reported a median age of 58 years for patients with liposarcomas [34]. However, a larger occurrence in men (65%) than in women (35%) was found in our cohort. Pleomorphic liposarcomas have an especially high rate of local recurrence and distance metastization of 35% [35]. Interestingly, in our study three of the eight under-diagnosed malignant tumors were located in the lower leg/foot region. All three were MXLPSs, and two of these were falsely diagnosed as neurinoma in the MRI report. Moreover, the average tumor mass of under-diagnosed malignant tumors measured by MRI was 139.9 cm^3^, almost five times less than the average tumor volume of liposarcomas measured by MRI. In summary, a small tumor located in a distal body region has to be given the same attention as larger tumors in more central body regions. 

LPS is a potentially lethal disease with poor overall survival. Three treatment options exist: surgery, chemotherapy, and radiotherapy. The only curative treatment is surgery with wide resection and tumor-free margins. RT only reduces local recurrence of an LPS, and chemotherapy is linked to limited overall survival [15,27]. However, this might be due to a negative selection of patients for chemotherapy, with a tendency to treat elderly patients with larger tumors in a palliative pattern (being solely treated with CTX). In advanced LPSs, cytotoxic CTX has shown response rates of only 10% [36,37]. Garbay and colleagues reported an improved overall survival of any CTX-treated patient when compared with the best supportive care [38]. 

After pathologic analysis, all cases with intermediate or malignant tumors should be discussed in an interdisciplinary tumor conference after a radiologic tumor staging of the patient [39]. In some cases, neoadjuvant chemotherapy can downsize the tumor to a resectable size. Complete resection should always be favored in malignant tumors. 

As a limitation of this study, 10 tumors of the 68 liposarcomas were excluded from measurement after resection due to neoadjuvant chemotherapy (*n* = 5). Five tumors were not resected after the biopsy. All general limitations of a retrospective analysis apply to this study. The resolution of the MR tomographs (slice thickness and magnetic field strength in tesla) was not considered since they were not completely reported. Surgical treatments were performed by a homogenous group of surgeons (three experienced surgeons at one institution); however, a heterogeneous group of radiologists (>20) and pathologists performed the imaging/tumor analyses.

## 5. Conclusions

Lipomatous tumors range from benign lipomata to highly malignant liposarcomas. MR imaging is currently the most precise diagnostic tool available; complemented with a contrast medium, an experienced radiologist can provide a valuable estimation of a tumor’s entity. The tumor’s size, location, and depth within the tissue; the homogeneity of the T2-weight signal; and the presence of septa or nodules should be considered. The radiologic report is of utmost importance for the further treatment of soft tissue tumors. This was shown in a large, single-center collectivity of patients. Our study reveals that 73.3% of lipomatous tumors were correctly estimated. However, 21.7% of the tumors were over-diagnosed. ALTs were significantly larger in size than lipomata and occurred in significantly older patients than lipomata. A threshold size or volume for safely differentiating an ALT from a lipoma cannot be proclaimed.

With further improvement in the quality of MRI, an experienced radiologist might be able to give an even more exact provisional diagnosis of a tumor. This could encourage surgeons to primarily resect some tumors safely without a prior biopsy. However, the quality of the provisional diagnosis by the radiologist depends on that individual’s experience with STT imaging. Therefore, our data should be an impetus for larger, prospective clinical trials. 

## Figures and Tables

**Figure 1 diagnostics-12-01281-f001:**
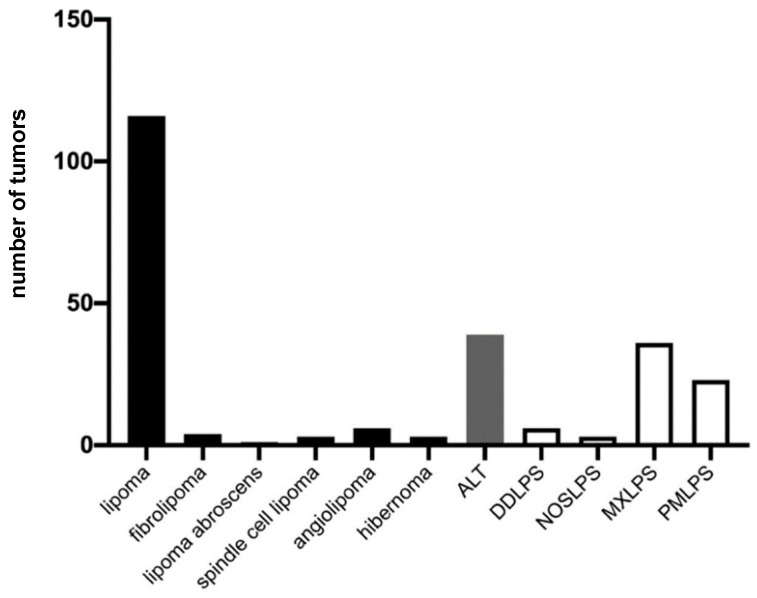
Tumor entities according to the final histologic diagnosis. Lipomatous tumors are further subclassified, and the number of subentities is depicted. Benign entities are depicted in black, intermediate in gray, and malignant tumors in white.

**Figure 2 diagnostics-12-01281-f002:**
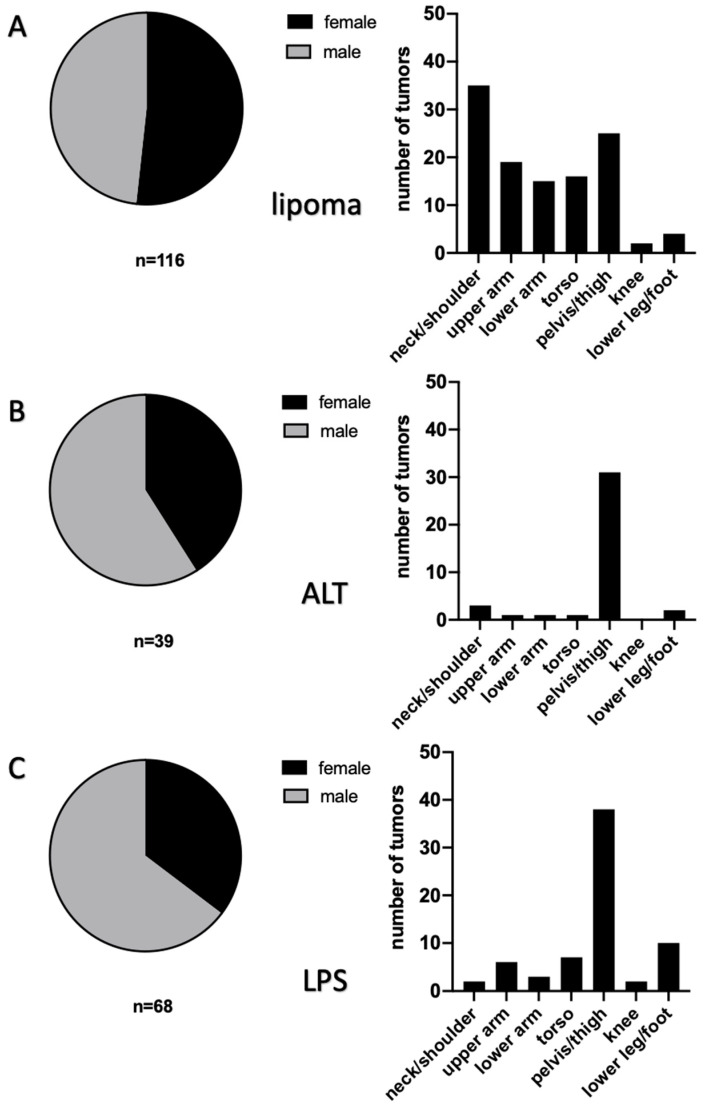
Gender distribution and body region distribution of lipomatous tumors. Section (**A**) of the figure gives an overview of gender and body region distribution of lipomata. Likewise, section (**B**) shows that more men than women were affected by ALTs, and by far, most ALTs were found in the pelvis/thigh region (*n* = 31). Section (**C**) depicts the results for LPSs. A total of 44 LPSs were identified in male patients, in contrast to 24 in females. The highest incidence per region was found in the pelvis/thigh region (*n* = 38).

**Figure 3 diagnostics-12-01281-f003:**
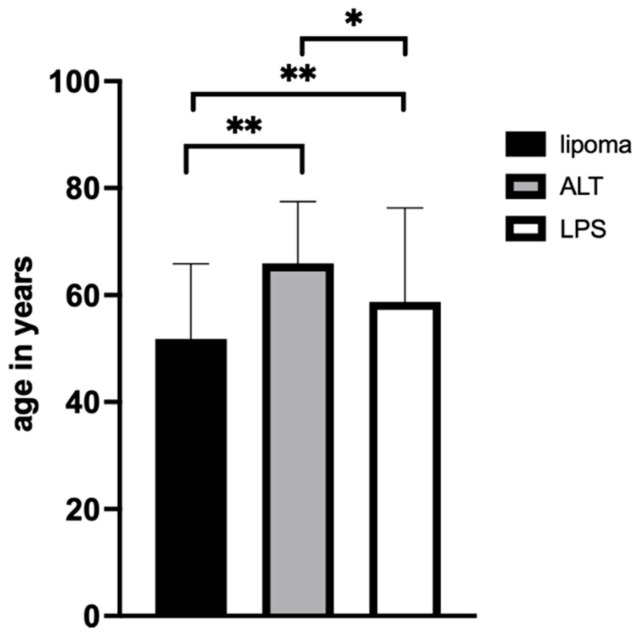
Age distribution of lipomatous tumors at the time of surgery. Patients with lipomas were on average 51.8 ± 14.0 years old. ALTs were more frequent in older people (mean 66.0 ± 11.5 years old). Thus, this cohort was significantly older than patients with lipomas (*p* < 0.0003). No significant difference was found in the mean age of patients with an LPS (*p* = 0.19). Patients with an LPS had an average age of 58.8 ± 17.5 years; they were significantly older than patients with a lipoma (*p* = 0.0146). Normal distribution was proven by a Shapiro–Wilk test. For 2-way statistical analysis, an ANOVA (CI = 95%) was performed. * indicates *p* < 0.05; ** indicates *p* < 0.005.

**Figure 4 diagnostics-12-01281-f004:**
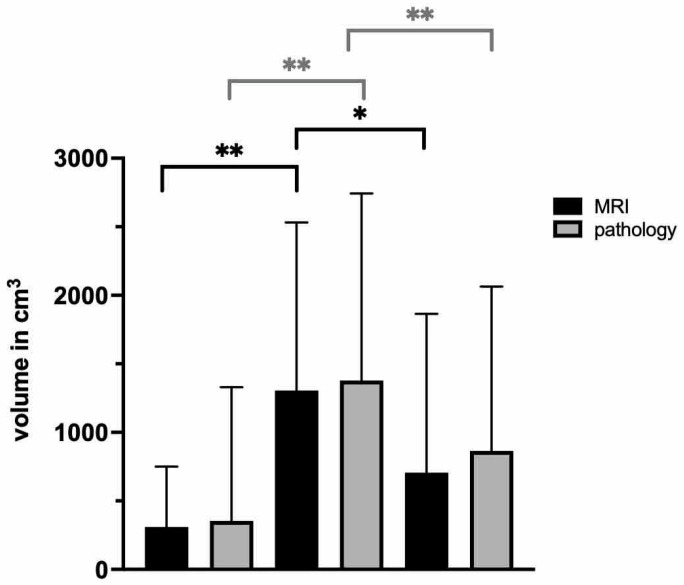
Volume differences between lipomatous tumors. The black columns depict the tumor volumes measured by MRI; the gray column depicts the volumes measured by the pathologist. * indicates *p* < 0.05; ** indicates *p* < 0.005.

**Table 1 diagnostics-12-01281-t001:** Entities of tumors under-diagnosed by MRI. In total, 12 tumors were under-diagnosed regarding their malignancy by MRI.

Histologic Entity	Frequent False Diagnosis by MRI	N	Total in Cohort	False in%	Total False Per Entity in%
ALT	Lipoma	4	39	10.3	10.3
MXLPS	Hemangioma	1	36	2.8	
MXLPS	Hemorrhage	1	36	2.8	
MXLPS	Neurinoma	2	36	5.6	
MXLPS	Lipoma	1	36	2.8	
MXLPS	Schwannoma	1	36	2.8	16.7
DDLPS	Hemangioma	1	6	16.7	16.7
PMLPS	Cyst	1	24	4.2	17.4

## Data Availability

The raw data are available upon reasonable request from the corresponding author.

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
