# Peer review of "Lipomatous Tumors: A Comparison of MRI-Reported Diagnosis with Histological Diagnosis"

_diagnostics, 2022, doi:10.3390/diagnostics12051281_

Round 1
Reviewer 1 Report
This manuscript provieds meaningful review and comparison of diagnosis of lipomatous tumors via MRI and histology.
This article emphasizes on importance of radiologic reports on soft tissue tumors which can be used in surgical treatment
along with size tendency of lipomatous tumors in elderly patients.
Only few things need revision.
1. Description of abbreviation ALT (atylpical lipomatious tumor) is missing in both abstract line 18 and introduction on line 41
2. Description of abbrevitation of LPS (liposarcomas) is missing in abstract line 18.
3. Line 162, mentioning that 73.3% of your study will be helpful for reviewers to comprehend better.
4. Line 154, citation for Coran et al is missing.
Reviewer 1 Report
This manuscript provides meaningful review and comparison of diagnosis of lipomatous tumors via MRI and histology.
This article emphasizes on importance of radiologic reports on soft tissue tumors which can be used in surgical treatment along with size tendency of lipomatous tumors in elderly patients.
Only few things need revision.
1. Description of abbreviation ALT (atypical lipomatous tumor) is missing in both abstract line 18 and introduction on line 41
2. Description of abbreviation of LPS (liposarcomas) is missing in abstract line 18.
3. Line 162, mentioning that 73.3% of your study will be helpful for reviewers to comprehend better.
4. Line 154, citation for Coran et al is missing.
Response to Reviewer 1
Dear Reviewer,
We are grateful for your comments, which will significantly improve our manuscript. Below in the text, we will answer to your comments point-by-point.
This manuscript provides meaningful review and comparison of diagnosis of lipomatous tumors via MRI and histology.
This article emphasizes on importance of radiologic reports on soft tissue tumors which can be used in surgical treatment along with size tendency of lipomatous tumors in elderly patients.
Only few things need revision.
1. Description of abbreviation ALT (atypical lipomatous tumor) is missing in both abstract line 18 and introduction on line 41.
• Amendments to the text have been made.
2. Description of abbreviation of LPS (liposarcomas) is missing in abstract line 18.
• The abbreviation ́s full meaning has been added to the text. Furthermore, we updated the abbreviation ́s list in Appendix A.
3. Line 162, mentioning that 73.3% of your study will be helpful for reviewers to comprehend better.
• “In our study” was added to emphasized that the result of 73.3% relates to our study.
4. Line 154, citation for Coran et al is missing.
• Thank you for the hint. The statement about the rareness and consequently difficult process of finding the right diagnosis was supported by the reference (17) Coran A.: Magnetic Resonance Imaging Assessment of Lipomatous Soft-tissue Tumors. In Vivo. 2017

Reviewer 2 Report
Thank you for the opportunity to revise the manuscript "Lipomatous Tumors: A Comparison of MRI-reported Diagnosis to Histological Diagnosis". The paper is a large case series of lipomatous tumors in which authors compared the report of MRI with the histological one.
The topic is interesting, however, not novel. The accumulation of patients does not improve what is known on the topic, nor might it lead to a change in practice.
Moreover, as listed by the authors, studies already published have concluded on the same topic with the same results. So the paper is unable to enhance the literature about the use of MRI in the differential diagnosis of lipomatous tumors.
Author Response
Reveiwer 2 Report
Thank you for the opportunity to revise the manuscript "Lipomatous Tumors: A Comparison of MRI-reported Diagnosis to Histological Diagnosis". The paper is a large case series of lipomatous tumors in which authors compared the report of MRI with the histological one. The topic is interesting, however, not novel. The accumulation of patients does not improve what is known on the topic, nor might it lead to a change in practice. Moreover, as listed by the authors, studies already published have concluded on the same topic with the same results. So the paper is unable to enhance the literature about the use of MRI in the differential diagnosis of lipomatous tumors.
Response to Reviewer 2
Dear Reviewer,
We thank you for reviewing our manuscript. Your evaluation of our work is valuable to us.
The result was a correctness of the provisional diagnosis in MRI to the histological diagnosis in 73.3% of the cases. Furthermore, we could show that the number of underdiagnosed tumors is by far smaller (5.0%). Data in the available literature is fairly scare. We analyzed 240 tumors. Similar studies have
analyzed their data in smaller cohorts: Coran et al. (n=54), O ́Donnell et al. (n=60) Leporq et al. (n=81), Brisson et al. (n=87) Gelineck et al. (n=43) Evans et al. (n=61). Downes et al. (n=19).
The quality of medical imaging and especially MRI, improves constantly. MRI scanners are fitted with stronger magnetic fields and the signal detectors improve. New scanning sequences are under development and in the future machine intelligence might support the radiologist in the process of
finding the right diagnosis.
Our retrospective study gives a contemporary overview of the correctness of MRI-reports, which were created during the day-to-day routine.
We agree with you, that our results do not revolutionize the process of soft tissue tumor diagnostics.
However, we still believe that our analysis adds important information to current knowledge and inspires researchers to further investigate imaging of soft tissue tumors.

Reviewer 3 Report
- study's idea is interesting
- the authors should refrain from using abbreviations in the abstract
- the authors should explain all the abbreviations used before using them
- please edit the text thoroughly for the English language
- no treatment decision is ever made based only on imaging findings, so the authors should rephrase this idea, repeated more than once in the paper's text ("Whether the treatment of a STT becomes an aesthetic or oncologic is usually depends on the radiologist's assessment")
Reviewer 3 Report
Suggestions for Authors
• study's idea is interesting
• the authors should refrain from using abbreviations in the abstract
• the authors should explain all the abbreviations used before using them
• please edit the text thoroughly for the English language
• no treatment decision is ever made based only on imaging findings, so the authors should rephrase this idea, repeated more than once in the paper's text ("Whether the treatment of a STT becomes an aesthetic or oncologic is usually depends on the radiologist's assessment")
Response to Reviewer 3
Dear Reviewer,
We appreciate your effort to review our manuscript and we are grateful for your suggestions. Below, all your concerns are replied by us one-by-one:
• the authors should refrain from using abbreviations in the abstract
• Amendments have been made to the abstracts.
• the authors should explain all the abbreviations used before using them
• The entire manuscript was searched for abbreviations. We ensured that they were used in full written words, when used for the first time. Furthermore, the Appendix A with the list of abbreviations was update.
• please edit the text thoroughly for the English language
• The manuscript was initially edited by a professional English Editing service
(American Manuscript Editors). However, during the process of internal revisions
some errors might have occurred. Therefore, we checked the manuscript with the GRAMMERLY software to improve the language and style.
• no treatment decision is ever made based only on imaging findings, so the authors should rephrase this idea, repeated more than once in the paper's text ("Whether the treatment of a STT becomes an aesthetic or oncologic is usually depends on the radiologist's assessment")
• We totally agree with you, that a treatment decision cannot be made solemnly based on MRI. In concordance with your suggestion the sentence was rephrased to: After clinical examination of the swelling and conduction of an MRI, the radiologist’s assessment frequently decides to which specialist the patient is referred to by a general practitioner. The speed of the diagnostic process is especially crucial in malignant STTs. (ll. 186)

Round 2
Reviewer 3 Report
This reviewer would like to thank the authors for their responses; their attempt to improve the submission is laudable. Please edit the paper for the English language once more; there are still some things to correct (for example, "seize" should be "size," etc.). Also, please expand the "Conclusions" section with some directions for the future.
Author Response
Reviewer´s Report:
This reviewer would like to thank the authors for their responses; their attempt to improve the submission is laudable. Please edit the paper for the English language once more; there are still some things to correct (for example, "seize" should be "size," etc.). Also, please expand the "Conclusions" section with some directions for the future.
Authors´ Response:
Dear Reviewer,
we are grateful for your effort to review our manuscript once-again. The paper was sent to the publishers-own language editing service to improve the English language in the text. As you requested, in the revised conclusion section we give the reader an perspective for the future of soft tissue tumor imaging.
